# Quantitative Assessment of the State of Threat of Working on Construction Scaffolding

**DOI:** 10.3390/ijerph17165773

**Published:** 2020-08-10

**Authors:** Marek Sawicki, Mariusz Szóstak

**Affiliations:** Department of Building Engineering, Faculty of Civil Engineering, Wroclaw University of Science and Technology, 27 Wybrzeże Wyspiańskiego st., 50-370 Wrocław, Poland; marek.sawicki@pwr.edu.pl

**Keywords:** scaffolding, occupational accidents, occupational safety, hazards at workplaces, construction industry

## Abstract

Working at height, and especially on construction scaffolding, is one of the most accident-prone situations on a construction site. The article attempts to assess the state of threat of working on scaffolding on the basis of the proposed coefficients concerning the possibility of an occupational accident occurring. The article presents the analysis of 10 parameters, which were classified into three groups of factors that cause accidents: technical, organizational, and human factors. In order to assess the state of threat of working on scaffolding, partial hazard factors and a simplified and accurate factor of the state of threat of working were proposed. The coefficients were determined on the basis of the data collected from post-accidental control reports on occupational accidents occurring on scaffolding in the construction industry that took place in Poland in five voivodeships in the years 2008–2017, and also on the basis of the obtained results of research on 120 scaffoldings conducted in the years 2016–2018. Based on the determined factors, it was possible to determine the probability of an undesirable event, in other words, an occupational accident. In addition, the developed test method proposed numerical scales for assessing the state of threat of working on scaffolding. The form proposed in the article for assessing the state of threat of working on scaffolding, which was developed using a spreadsheet, can provide support for people managing work at workstations involving scaffolding, for example, construction directors, construction engineers, work managers, or construction managers.

## 1. Introduction

The construction industry is one of the most dangerous industries in many parts of the world, as evidenced by the data published by global health and safety organizations [1,2,3]. The conducted review of the subject literature shows that the most common event that is incompatible with an appropriate conduct of the work process is a fall from height [4]. Research carried out by Chi and Han showed that occupational accidents most often occur as a result of: falling from height or a fall at the same level (43.9%), being hit by a moving object (25.7%), being trapped under something or crushed (10.0%), and affected by electric shock (6.1%) [5]. On average, every third occupational accident in the construction industry is caused by falling from height, and the main causes include unsafe working conditions, a lack of protective barriers, and also improper or a lack of use of personal protective equipment [6]. For example, every year in Japan about 300 employees die in occupational accidents in the construction industry, with 134 accidents being caused by falling from height [7]. Accidents, such as slips and falls, have higher prevalence for older workers than any other type of accidents. This is due to the fact that the older workers are more vulnerable than younger workers in the workplace [8].

Despite the noticeable improvement in safety at construction sites and a significant reduction in the overall number of accidents in the construction industry, a fall from height still causes more than 47% of all fatalities in this sector [9]. Working at height includes activities carried out on scaffolding, ladders, columns, masts, tower structures, and chimneys [10]. The most common fatal accidents involve falls from height from: scaffolding (24%), building girders or other steel constructions (19%), floors to unprotected openings (14%), ladders (12%), and roof edges (7%) [11]. Research conducted by Rubio-Romero et al. shows that falls from height to a lower level occur in more than 55% of all accidents, and construction scaffolding is one of the most common material factors associated with the fall of an injured person [12]. According to the statistical data published by the Central Statistical Office in Poland, a fall from scaffolding constitutes over 38% of all falls from height, which means that every third fall from height occurred from scaffolding [13].

Scaffolding is a temporary building structure that enables work to be carried out at a height ranging from two to several dozens of meters above the ground. Scaffoldings are used during the construction of new buildings, repairs, and modernization of existing buildings. The basic features that characterize scaffolding include: dimensions (size), height, span width and length, maximum load on a working platform, maximum height of the last working platform, foundations, load-bearing capacity of the ground, a support’s load bearing capacity, and also the method and location of anchoring [14].

Research shows that the main cause of falls from height involves the loss of an employee’s balance on scaffolding [15] and the falling of a person off scaffolding through the free space between the handrail and the working platform [16]. Falls from height most often result in the death of the victim [17], and only in a small number of cases does the fall only lead to serious injuries [18].

Identification and assessment of hazards and risks is a necessary step in safety management. Analysis and risk assessment is performed to identify and assess the level of risks at workplaces. Occupational risk is the possibility of adverse events related to the work being performed causing losses [19]. The occupational risk analysis encompasses different types of risks faced by employees in the workplace, such as physical, mechanical, chemical, biological, and ergonomic [20].

The purpose of occupational risk assessment is to identify potential sources of hazards and to conduct appropriate corrective or preventive actions to eliminate the possibility of an accident at work or occupational disease. The risk analysis methods encompass different types of qualitative and quantitative risk analysis. These methods differ from each other in: the procedure when collecting information about risk, resource of information necessary to assess, and criteria used in assessment. The criteria for the division of occupational risk assessment methods into qualitative and quantitative might be access to the statistical data. Quantitative estimation is carried out only when there is access to an adequate amount of compliant statistical data regarding: the number and types of accidents at work, the time of exposure to the work environment, et cetera. In the absence of access to numerical data (quantities) allowing for precise risk estimation, qualitative methods are used. In these methods the estimated effects of the event and the probability or frequency of occurrence of certain effects are attributed to conventional qualities characterizing their value. The assigned qualities can be descriptive or measured by the quality of something rather than its quantity [21]. The choice of the methods depends on the type of the assessed work process, the type of hazards, and availability of data.

For instance, job safety analysis (JSA) also known as job hazard analysis (JHA) is currently one of the most important onsite risk management methodologies. JSA is a practical method for identifying, evaluating, and controlling risks in industrial procedures [22,23]. The differences between construction sites and manufacturing facilities give rise to the need for a specialized method for construction and therefore the dedicated method has been developed for the construction industry—construction job safety analysis (CJSA) [24]. Unfortunately, both aspects of this method tend to fail at identifying all the potential hazards involved in a job and can lead to accidents. Therefore, a new method of energy source based job safety analysis (ESBJSA) was developed to identify all the potential hazards in a specific job or task [25]. The classic methods for identification of hazards use a checklist. Based on the checklist it is possible to indicate the risk level [26]. All of these methods use the qualitative method and the obtained assessment is subjective. That is why quantitative methods are sought. Only quantitative methods allow for accurate quantitative occupational risk assessment.

The purpose of the research was to develop a methodology for quantitatively assessing the state of threat of working at workplaces that use scaffolding. This methodology will be based on the calculated indicators for the assessment of technical, organizational, and human factors. To quantitatively assess the state of threat of working on scaffolding, partial coefficients of the possibility of a dangerous situation occurring were proposed, as well as a simplified and accurate risk assessment coefficient.

## 2. Materials and Methods

The article presents the analysis of 10 parameters, which were classified into three groups of factors that cause accidents: technical, organizational, and human [27]. Technical factors are the result of incorrect and defects of material factors and include the type of scaffolding where the accident occurred (*t*), the technical condition of the scaffolding—damage (*s*), and foundations (*p*). Organizational factors are related to the inappropriate general organization work of construction enterprises and the inappropriate organization of a workplace and involve the type of jobs (*r*), the height from which the injured person fell (*h*), the time when the accident occurred (*g*), and the size of the enterprise where the accident occurred (*f*). Human factors are directly related to employees at workplaces, with knowledge of hazards, regulations and rules of occupational health and safety and include the age of the injured person (*w*), experience with regards to working on scaffolding (*d*), and also employment status (*z*).

### 2.1. Accident Analysis

In the conducted research, the main sources of the data on occupational accidents are archival documents developed by inspectors of the National Labour Inspectorate in Poland. As a result of post-accidental proceedings, a labor inspector prepares a control protocol that contains information on the circumstances and causes of the investigated event, especially: information about the time and place of the occurrence, performed activities, the data about the injured person, and also a description of the cause of the accident. The preliminary analysis of the collected data, and also literature studies carried out by the authors of the article, showed the existence of a relationship between the occurrences described in the control protocols and the physiological parameters of the human body that change in subsequent periods of work and life [28]. In the conducted research, 219 people injured in occupational accidents involving scaffolding that occurred in five Polish voivodeships within the years 2008–2017 were analyzed. The voivodeship is the highest-level administrative subdivision of Poland, corresponding to a “province” in many other countries [29].

A partial hazard coefficient was proposed for the analysis of the following 8 parameters: the type of scaffolding on which the accident occurred (*t*), the type of work (*r*), the height from which the injured person fell (*h*), the time of the accident when the accident occurred (*g*), the size of the enterprise where the accident occurred (*f*), the age of the injured person (*w*), the experience of people working on scaffolding (*d*), and employment status (*z*). It was determined using the following formula:(1)$$W_{j}=\frac{i_{j}}{l_{j}},$$
where:

*W_j_* is the partial hazard coefficient for the analyzed parameter, *j* = (*t,r,h,g,f,w,d,z*);

*i_j_* is the number of people injured for the analyzed *j*-th parameter; and

*l_j_* is the number of all victims (∑i) for the analyzed *j*-th parameter.

In the above formula, partial hazard coefficients refer to the number of all analyzed occupational accidents.

### 2.2. Construction Scaffolding Tests

The second sources of data were the results obtained within the framework of the scientific and research project implemented in Poland in the years 2016–2018 entitled “Model of the assessment of risk of the occurrence of building catastrophes, accidents and dangerous events at workplaces with the use of scaffolding” (ORKWIZ). As part of this project, 120 scaffoldings were tested. Façade scaffoldings that form surfaces next to building objects, with a scaffolding area from 50 to 1500 m^2^ and a height of up to 15 storeys, were tested. The scope of the conducted research included the implementation of the following: scaffolding inventory, damage inventory, load inventory, and checking of anchoring [30].

Part of the ORKWIZ research project involved the identification of damage, which concerned such elements as: frames, handrails, working platforms, toe boards, and communication lines—ladders. The main damage of frames included the bending of the stand in the frame plane, the bending of the stand out of the frame plane, and the bending of the frame’s stand pipe. Another group of damage concerned handrail damage that occurred in the form of bends out of the plane or within the scaffolding plane. For working platforms, the most common damage included bending along the platform or bending in its cross-section [31]. The value of the partial hazard coefficient ($W_{s}$) was determined using the following formula:(2)$$W_{s}=\frac{l_{ue}}{l_{e}}$$
where:

*l_ue_* is the number of damaged elements (frames, working platforms, handrails, toe boards, ladders); and

*l_e_* is the number of all elements (frames, working platforms, handrails, toe boards, ladders).

Another element was verification of the correctness of the foundation method [32]. Construction scaffolding should be placed on stabilized ground. When scaffolding is mounted to the ground, additional elements are used, for example, underlays. The underlays should be laid on the prepared ground, perpendicular to the wall of a building structure, and in a way that ensures pressure against the ground with the whole bottom plane of the underlay. It is unacceptable to place frames on cracked and broken underlays, on wedge underlays, or those made of brick. The value of the partial hazard coefficient (*W_p_*) was determined using the following formula:(3)$$W_{p}=\frac{l_{up}}{l_{p}}$$
where:

*l_up_* is the number of incorrectly set frames, damaged underlays; and

*l_p_* is the number of vertical elements of all frames, foundation pairs.

### 2.3. Quantitative Assessment

In order to assess the state of threat of working on workplaces that use scaffolding, partial coefficients of the possibility of a dangerous situation occurring on a given scaffolding, and also a simplified and accurate coefficient of the state of threat of working on this scaffolding, were proposed. A simplified coefficient for assessing the state of threat of working on scaffolding (*UW_z_*) is obtained by summing the individual partial hazard coefficients using the following formula:(4)$$UW_{z}=\sum W_{j}$$
where:

*UW_z_* is a simplified coefficient of assessing the state of threat of working on scaffolding; and

*∑W_j_* is the sum of partial hazard coefficients *j* = (*t,s,p,r,h,g,f,w,d,z*).

In order to assess the state of threat based on a simplified coefficient of assessing the state of threat of working on scaffolding (*W_z_*), a scale from 0 to 10 was proposed, in which:the value of 0.1 means that the probability of a hazard occurring is theoretically possible and equal to 0.000001 (0.0001%);the value of 0.2 means that the probability is theoretically possible and equal to 0.00001 (0.001%);the value of 0.5 means that the probability is possible and equal to 0.0001 (0.01%);the value of 1.0 means that the probability is possible and equal to 0.001 (0.1%);the value of 3.0 means that the probability is possible and equal to 0.01 (1%);the value of 6.0 means that the probability is possible and equal to 0.1 (10%); andthe value of 10.0 means that is very possible and equal to 0.5 (50%).

The exact coefficient for assessing the state of threat of working on scaffolding (*W_z_*) is obtained by calculating the weighted average of the selected (for technical and organizational factors) or calculated (for human factors) partial hazard coefficients with specific weights, using the following formula:(5)$$W_{z}=\frac{W_{T}\cdot x_{T}+W_{O}\cdot x_{O}+W_{H}\cdot x_{H}}{x_{T}+x_{O}+x_{H}}$$
where:

*W_z_* is the overall hazard coefficient;

*W_T_* is the summed partial hazard coefficient for technical factors;

*W_O_* is the summed partial hazard coefficient for organizational factors;

*W_H_* is the summed partial hazard coefficient for human factors;

*x_T_* is technical factor with weight equal to 0.25;

*x_O_* is organizational factor with weight equal to 0.48; and

*x_H_* is human factor with weight equal to 0.27.

The weights of individual coefficients were determined on the basis of previous studies regarding the analysis and assessment of the causes of accidents involving construction scaffoldings. Based on the analysis of 177 accidents that occurred in workplaces where scaffolding was used, 1132 causes of accidents were identified. Technical factors constituted 24.6%, organizational factors 48%, and human factors 27.4% of all identified causes [33]. Based on this data, the following weights were determined: for technical factors—0.25, for organizational factors—0.48, and for human factors—0.27.

The maximum achievable exact value of the hazard coefficient of assessing working on scaffolding is equal to 3.5. For the assessment of the state of threat of working on scaffolding, the following ranges, corresponding to the respective groups of hazards, were proposed:0 < *W_z_* < 1.1: there is almost no threat,1.2 < *W_z_* < 2.1: a threat is insignificant,2.1 < *W_z_* < 2.8: a threat exists,2.9 < *W_z_* < 3.1: a threat occurs to a large extent, and3.2 < *W_z_* < 3.5: a threat occurs to a high degree.

## 3. Results

### 3.1. Accident Analysis

The results of the analyses of the tested parameters of occupational accidents involving construction scaffolding were obtained as a result of the analysis of 219 people injured in occupational accidents. In the conducted analyses, the size of the research group changed due to a lack of data in some control protocols. Therefore, for some analyses, the accidents that did not have the analyzed parameters were rejected.

#### 3.1.1. Scaffolding Type (*t*)

Table 1 presents the number of injured people with regards to the type of scaffolding on which the accident occurred, and also the partial hazard coefficient *W_t_*.

Based on the obtained data, occupational accidents most often occurred on Warsaw-type scaffolding (*W_t_* = 0.34) and on frame-type scaffolding (*W_t_* = 0.33). From the obtained values *W_t_*, it can be seen that Warsaw- and frame-type scaffoldings are constructions on which accidents occur more frequently than on other types of scaffolding. The Warsaw-type scaffolding is the columnar scaffoldings. It can be used both inside and outside buildings. The Warsaw-type of scaffold is characterized by simple construction, easy, and quick montage. The elements are connected without screws, with usage of tenon joints [34]. Figure 1 shows an example of the Warsaw-type scaffolding.

When comparing the scope of performed work and the size of frame-type scaffolding, it can be assumed that the number of accidents on this type of scaffolding should be higher. However, Warsaw scaffoldings are usually used on small construction sites, where conducted work is influenced by an economic system. This often translates into lower health and safety awareness. The obtained results confirm that the general safety level of standardized scaffolding is higher than that of non-standardized scaffolding [35].

#### 3.1.2. Type of Conducted Work (*r*)

Table 2 contains the data on the type of conducted work during which the accident occurred and also the partial hazard coefficient *W_r_*.

Occupational accidents involving scaffolding most often occurred during the erection of new buildings (*W_r_* = 0.50). The second most accident-prone jobs were renovation works (*W_r_* = 0.38). The lower value of the coefficient results from a smaller number of conducted renovation work in relation to work related to the erection of new buildings. It was during these construction works that occupational accidents most often occurred.

#### 3.1.3. Height from which the Injured Person Fell (*h*)

Table 3 provides analysis of the relationship of the number of injured people with regards to the height from which the injured person fell, and also the partial hazard coefficient *W_h_*.

Data analysis indicates that as the height at which construction works are carried out increases, the effect of the accident becomes more severe. Therefore, the theory that the number of accidents with fatal and severe effects increases with the height at which employees are working, the occurrence of a light accident decreases with height, and also at above 8 m there are usually no light accidents, is confirmed. Moreover, with the increasing altitude at which construction works are carried out physiological responses are changed. Some easy tasks at ground level become more difficult when performed in high-level workplaces [36]. The highest value of the partial hazard coefficient *W_h_*, equal to 0.37, occurred for accidents that occurred from a height of 2–4 m, in other words, from the first level of scaffolding. Most often, falling from construction scaffolding resulted in severe injuries (111 injured people, which constitutes over 50% of all occupational accidents).

#### 3.1.4. Time when the Accident Occurred (*g*)

Table 4 presents the number of injured people with regards to the time when the occupational accident occurred, and also the partial hazard coefficient *W_g_*.

For the analyzed data, the highest value of the partial hazard coefficient (*W_g_*) was recorded for two time intervals, namely, between 09:00 and 09:59 (value of 0.14) and between 14:00 and 14:59 (value of 0.15). This is due to the partial relaxation of employees just before the breakfast break and also before leaving work. The finding is similar to that in previous studies [28,37,38].

#### 3.1.5. The Size of the Enterprise Where the Accident Occurred (*f*)

Table 5 presents the data on the number and size of construction enterprises where the accident occurred, the number of people working in these enterprises, and also the partial hazard coefficient *W_f_*.

Most people suffered accidents at workplaces involving scaffolding in micro-enterprises that employ 1–9 people. The number of injured people in this group of enterprises amounted to 129, which represents 59% of all injured people in all types of construction enterprises. The second largest group are small enterprises that employ 10–49 employees.

The obtained data indicate that a particularly high accident rate occurs in micro- and small enterprises, which in the structure of Polish construction companies constitute over 96% of all construction enterprises [39]. The reasons for the increased number of occupational accidents in these enterprises are mainly due to: the neglecting of OHS rules by employees, rushing and organizational chaos, high employee turnover, a lack of or improper application of collective protection measures, et cetera. Similar conclusions were obtained in the conducted analysis of accidents at small construction enterprises in Taiwan. Based on the obtained results, it was found that the health and safety management is less adequate for small construction enterprises compared to that of large construction enterprises [40].

#### 3.1.6. Age of the Victim (*w*)

Table 6 shows the distribution of the number of victims with regards to the age of the injured person, and also the partial hazard coefficient *W_w_*.

The determined values indicate that a higher value of the determined partial hazard coefficient (*W_w_*) corresponds to a greater danger of working on scaffolding and a higher possibility of an occupational accident occurring.

It is surprising for the authors that the number of victims in the 18–19 age group was equal to 3. Based on the analysis of literature, a higher amount of victims were expected to be found in this group. According to the authors, the largest number of accident events in the age group 40–49 is related to the routine attitude of employees to their professional activities (often with nearly 25 years of professional experience), as well as to a decrease in psychomotor skills at this age. In the “over 60” age group, the observed decrease in the number of accidental events may be related to, among others, the changing of job post, retirement, or greater assurance at work [41,42].

#### 3.1.7. Experience of People Working on Scaffolding (*d*)

Table 7 presents the number of people injured in accidents with regards to their work experience in the company, and also the partial hazard coefficient *W_d_*.

Sixty-nine people were injured in occupational accidents that occurred at workplaces involving scaffolding in the first year of their work, contributing to 59% of all injured people. It is noticeable that longer work experience at a certain workplace corresponds to a lower number of accidents. A detailed analysis of the collected data indicates that people with low work experience are most frequently involved in accidents, even on their first day/week of work. This is mainly due to the lack of, or improper, training in the field of occupational health and safety; admission to work of an employee with medical contraindications or without medical examinations; and insufficient professional preparation of an employee [43].

#### 3.1.8. Employment Status (*z*)

Table 8 presents the number of people injured in occupational accidents involving scaffolding with regards to their employment status, and also the partial hazard coefficient *W_z_*.

The conducted analysis showed that the most frequently injured victims were employed under a fixed-term employment contract—65 persons, which constitutes 43% of all injured people. A large group of injured people are also workers employed under an indefinite employment contract (42 injured people) and also under civil law contracts, in other words, a contract of mandate (26 injured people) and a specific work contract (12 people). Unfortunately, construction practice shows that people employed under civil law contracts are not properly prepared to perform work, in other words, they did not undergo general and on-the-job training, did not have medical examinations, and do not possess current medical certificates concerning the absence of contraindications to work on a specific job position, for example, an employee is prohibited from lifting loads over 5 kg or working at heights, et cetera.

### 3.2. Construction Scaffolding Tests

The results obtained from the tests on 120 scaffoldings are presented below.

#### 3.2.1. Technical Conditions of Scaffolding (*s*)

In order to determine the partial hazard coefficient (*W_s_*), each tested scaffolding was subjected to an inventory with regards to its damage. Only 6 scaffoldings out of 120 that were tested had no damage, while the remaining 114 scaffoldings had at least one damaged element, for example, a frame, working platform, handrail, platform, or ladder. The average number of incidents of damage in the analyzed set of scaffoldings was 20. Table 9 shows the calculated partial hazard coefficient *W_s_* with regards to the surface area of the tested scaffolding.

#### 3.2.2. Foundations (*p*)

In order to determine the values of the partial hazard coefficient (*W_p_*) for each of the tested scaffoldings, irregularities in the area of foundations and the damage of underlays were identified. Only 25 tested scaffoldings were correctly placed, with irregularities being identified in the remaining 95 cases. The most common irregularities include:the foundation of scaffolding frames on separated sleepers—42 scaffoldings;scaffolding frames placed too close or on the edge of the underlay—23 scaffoldings;underlays that are more narrow than required—21 scaffoldings;a lack of underlays—20 scaffoldings;underlays covered with soil or debris—14 scaffoldings;cracked underlays—2 scaffoldings; andunderlays made of unsuitable materials, for example, hollow bricks, bricks, loose boards—10 scaffoldings.

Each of the incorrectly placed scaffoldings had at least one of the abovementioned irregularities. The average number of incorrectly placed frames per scaffolding was equal to 4. Table 10 shows the calculated partial hazard coefficient *W_p_* with regards to the surface area of the tested scaffolding.

## 4. Quantitative Assessment of the State of Threat of Working on Scaffolding

The obtained results from the analyses, which were presented in Section 3, enabled a hazard assessment form for working on scaffolding to be developed with the use of the proposed partial hazard coefficient (*W_j_*), which allows for the assessment of the probability of an accident occurring on scaffolding (*W_z_*). In the proposed form, a scaffolding user, for example, a construction manager, can assess the safety conditions and work hazards on the used scaffolding.

For this purpose, in order to determine the overall hazard coefficient (*W_z_*) and the state of threat, the following need to be done:for technical factors (*W_T_*):
o scaffolding type: the scaffolding type that is used during the execution of work and the value of the partial hazard coefficient that corresponds to the indicated type of scaffolding should be selected (*W_t_*);o technical condition: the number of damaged, as well as all elements, should be determined in the following order: frames, working platforms, handrails, toe boards, and communication lines—ladders. The value of the partial hazard coefficient (*W_s_*) was determined using formula (2) presented in Section 2. In the case of a lack of the possibility of assessing the technical condition of these items, the following designated values should be used:
▪ for scaffolding with an area from 30 to 600 m^2^: *W_s_* = 0.07,▪ for scaffolding with an area from 600 to 900 m^2^: *W_s_* = 0.04, and▪ for scaffolding with an area above 900 m^2^: *W_s_* = 0.03.o the number of vertical elements of all frames: pairs of scaffolding foundations and the number of incorrectly placed frames should be specified. The value of the partial hazard coefficient (*W_p_*) should be determined using formula (3) presented in Section 2. In the case of a lack of the possibility of assessing the correctness of foundations, the following designated values should be used:
▪ for scaffolding with an area from 30 to 300 m^2^: *W_p_* = 0.57,▪ for scaffolding with an area from 300 to 900 m^2^: *W_p_* = 0.50, and▪ for scaffolding with an area above 900 m^2^: *W_p_* = 0.47.for organizational factors (*W_O_*):
o the type of conducted work during which scaffolding will be used and the value of the partial hazard coefficient (*W_r_*) that corresponds to this type of work should be selected;o the height at which work will be carried out at a construction site: the appropriate heights at which scaffolding work is planned and the sum of the corresponding values of the partial hazard coefficient (*W_h_*) should be selected;o the time when scaffolding work will be carried out: the appropriate time periods in which carrying out work on scaffolding is planned and the summary values of the partial hazard coefficient (*W_g_*) that correlates to these periods should be selected; ando enterprise size: the appropriate value of the partial hazard coefficient (*W_f_*) that corresponds to the number of people employed and working on the analyzed scaffolding should be selected;
for human factors (*W_H_*):
o the age of people employed to work on scaffolding: the values of the partial hazard coefficient (*W_w_*) with regards to the number of employed people within age ranges should be calculated. The value of the partial hazard coefficient (*W_w_*) for individual age ranges should be determined using the following formula:
(6)$$W_{wn}=\frac{l_{on}\cdot U_{wn}}{l_{n}}$$
where:
*n* is the analyzed age range (18–19, 20–29, …, >60);*U_wn_* is the partial coefficient for the analyzed *n*-th age range;*l_on_* is the number of working people from the *n*-th analyzed age range; and*l_n_* is the total number of people working on the analyzed scaffolding;o work experience of people employed to work on scaffolding: should be calculated similarly to age as the value of the partial hazard coefficient (*W_d_*) with regards to the number of employed people with corresponding professional experience; ando employment status: should be calculated in the same way as age—as a partial hazard coefficient (*W_z_*) with regards to the number of employed people possessing an appropriate employment form.



## 5. Form of Quantitatively Assessing the State of Threat of Working on Construction Scaffolding

The form was developed in electronic version using MS Excel. Table 11 presents the proposed form of assessing the state of threat of working on scaffolding, together with an exemplary analysis of scaffolding.

The analysis of the assessment of the state of threat of working on scaffolding was carried out for scaffolding (shown in Figure 2) with the following parameters:a scaffolding area of approximately 1050 m^2^, number of modules 17, maximum number of working levels 11, width 4.50 m, and height 24.30 m,frame scaffolding,as a result of the damage inventory, the following were identified: 1 damaged frame out of 234 of such elements, 3 damaged working platforms out of 204 of such elements, 16 damaged handrails out of 856 of such elements, 1 damaged toe board out of 228 of such elements and no damage in communication elements,foundation analysis showed that 3 out of 18 underlays were incorrect—2 underlays were covered, 1 underlay was cracked,scaffolding was used next to a newly erected building,work carried out along the entire scaffolding height—above 12 m,work carried out from 07:00 to 17:00—a 10-hour day shift,15 people working on scaffolding, who are characterized by the following information:

o all people were employed for a definite period and on a full-time basis,o 4 people within the 40–49 age range and with experience ranging from 16 to 20 years,o 6 people within the 30–39 age range and with experience ranging from 6 to 10 years,o 5 people within the 20–29 age range and with lack of experience.

Based on the form of assessing the state of threat of working on the analyzed scaffolding, the obtained value of the simplified coefficient of assessing the state of threat for the scaffolding ($UW_{z}$) amounted to 4.07, which means that the probability of the occurrence of threat is theoretically possible $3.0\left(1\%\right)<UW_{z}=4.07<6.0\left(10\%\right)$ and equal to 0.04 (4%). The obtained value of the exact coefficient of assessing the state of threat for the scaffolding amounted to 1.68, which means that the threat occurs to a small extent ($1.2<W_{z}=1.68<2.1$).

## 6. Discussion

The conducted analysis of the data of people injured in occupational accidents involving scaffolding that took place in the construction industry in five voivodeships in Poland in the years 2008–2017, and also studies of 120 scaffoldings that were conducted in the years 2016–2018 as part of the project “Model of the assessment of risk of the occurrence of building catastrophes, accidents and dangerous events at workplaces with the use of scaffolding” (ORKWIZ) enabled a method for assessing the state of threat of working at workplaces using scaffolding to be developed.

In order to assess the state of threat of working on construction scaffolding, partial hazard coefficients and a simplified and accurate coefficient of the state of threat of working were proposed. Based on the determined coefficients, it is possible to determine the probability of an undesirable event, in other words, an occupational accident.

In identifying and quantitatively assessing the state of threat of working at workplaces it is of great importance to facilitate the risk assessment process in construction projects because risk assessment is a requirement in most legislations and safety standards. A great majority of small and medium construction enterprises are not familiar with risk assessment concepts and methods. In particular, small and medium enterprises are very likely to have difficulty finding the qualified personnel or time to carry out a proper risk assessment. The proposed methodology in this study introduces a new method of risk assessment, replacing the traditional methods and allows an easy and affordable way for small and medium enterprises to perform occupational risk assessment and analysis. The proposed risk assessment method introduces a powerful and practical control level strategy which would develop a safer, healthier, and more competitive workplace for small and medium construction enterprises.

## 7. Conclusions

The form proposed in the article for assessing the state of threat of working on scaffolding, which was developed using a spreadsheet, can provide support for people managing work at workstations involving scaffolding, for example, construction directors, construction engineers, work managers, or construction managers.

The inserted intermediate data of form for the state of threat of working on scaffolding and the obtained results (the simplified and exact coefficient for assessing the state of threat of working on scaffolding) indicate the elements that significantly contribute to employees’ accidents at work. The obtained results will allow appropriate preventive actions, which aim to improve work safety, to be formulated. All participants of the investment process: workers, but also construction site managers and supervisors, should be the recipients of these activities, who are also exposed to hazards and may suffer from accidents while performing their activities at a construction site.

Finally, according to the authors it is most important that the quantitative assessments of the state of threat of working on construction scaffolding conducted on the basis of the proposed form allows the estimating of the probability of the occurrence of hazards and allows predicting the possibility of occurrence of the dangerous events. Parameters and coefficients described by the authors allow for the comprehensive assessment of hazards and probability of accident occurrences.

## Figures and Tables

**Figure 1 ijerph-17-05773-f001:**
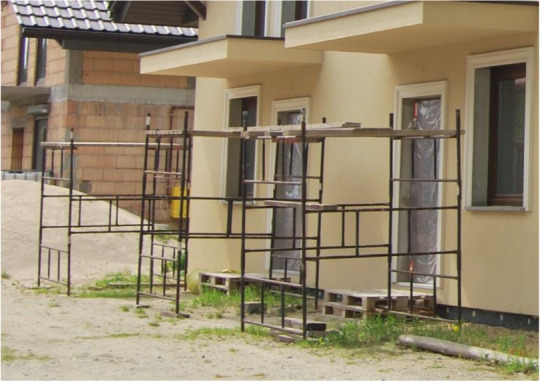
The example of the Warsaw-type scaffolding (author’s archive).

**Figure 2 ijerph-17-05773-f002:**
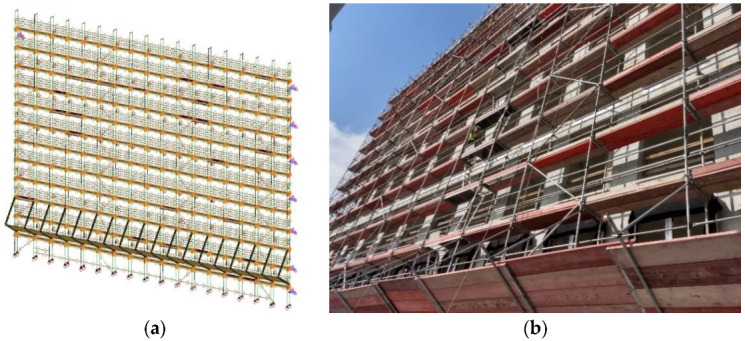
Scheme (**a**) and view (**b**) of the analyzed scaffolding.

**Table 1 ijerph-17-05773-t001:** Scaffolding type.

Scaffolding Type	System	Frame	Modular	Tubular	Carriageable	Warsaw	Suspended
Number of injured people ($i_{t}$)	31	64	3	5	24	65	2
Partial hazard coefficient ($W_{t}$)	0.16	0.33	0.02	0.03	0.12	0.34	0.01

**Table 2 ijerph-17-05773-t002:** Type of conducted work.

Type of Conducted Work	Construction Work	Renovation Work	Scaffolding Assembly	Scaffolding Dismantling	Total
Number of victims ($i_{r}$)—total	110	83	12	14	219
Number of victims ($i_{r}$)—fatal	22	15	1	3	41
Number of victims ($i_{r}$)—severe	56	42	6	7	111
Number of victims ($i_{r}$)—light	32	26	5	4	67
Partial hazard coefficient ($W_{r}$)	0.50	0.38	0.06	0.06	1.00

**Table 3 ijerph-17-05773-t003:** Height from which the accident occurred.

Height from which the Injured Person Fell	Number of Victims $\left(i_{h}\right)$—Total	Number of Victims $\left(i_{h}\right)$—Fatal	Number of Victims $\left(i_{h}\right)$—Severe	Number of Victims $\left(i_{h}\right)$—Light	Partial Hazard Coefficient $\left(W_{h}\right)$
less than 1 m	8	0	1	7	0.04
1–2 m	18	1	11	6	0.08
3–4 m	80	5	43	32	0.37
5–6 m	42	5	18	19	0.19
7–8 m	23	7	13	3	0.11
9–10 m	15	8	7	0	0.07
11–12 m	12	6	6	0	0.05
more than 12 m	21	9	12	0	0.10

**Table 4 ijerph-17-05773-t004:** Time of accidents.

Time when the Accident Occurred	07:00–07:59	08:00–08:59	09:00–09:59	10:00–10:59	11:00–11:59	12:00–12:59	13:00–13:59	14:00–14:59	15:00–15:59	16:00–16:59	17:00–17:59	18:00–06:59
Number of victims ($i_{g}$)	12	19	26	17	16	18	13	27	19	11	3	4
Partial hazard coefficient ($W_{g}$)	0.06	0.10	0.14	0.09	0.09	0.10	0.07	0.15	0.10	0.06	0.02	0.02

**Table 5 ijerph-17-05773-t005:** Enterprise size.

Enterprise Size (Number of Employees)	Micro-Enterprises (1–9)	Small Enterprises (10–49)	Medium Enterprises (50–249)	Big Enterprises (above 250)
Number of victims ($i_{f}$)	129	78	10	2
Partial hazard coefficient ($W_{f}$)	0.59	0.36	0.05	0.01

**Table 6 ijerph-17-05773-t006:** Age of victim.

Victim’s Age	18–19	20–29	30–39	40–49	50–59	>60
Number of victims ($i_{w}$)	3	33	30	35	41	6
Partial hazard coefficient ($W_{w}$)	0.02	0.22	0.20	0.24	0.28	0.04

**Table 7 ijerph-17-05773-t007:** Victim’s work experience.

Work Experience	1 Year and Less	2 to 3 Years	4 to 5 Years	6 to 10 Years	11 to 15 Years	16 to 20 Years	Over 20 Years
Number of victims ($i_{d}$)	69	20	10	11	2	2	2
Partial hazard coefficient ($W_{d}$)	0.59	0.17	0.09	0.09	0.02	0.02	0.02

**Table 8 ijerph-17-05773-t008:** Employment status.

Employment Status	Self Employed	Worker Employed for an Indefinite Period	Worker Employed for a Fixed Term	Trainee /s\Student	Specific Work Contract	Contract of Mandate
Number of victims ($i_{z}$)	5	42	65	1	12	26
Partial hazard coefficient ($W_{z}$)	0.03	0.28	0.43	0.01	0.08	0.17

**Table 9 ijerph-17-05773-t009:** Damage with regards to the scaffolding surface area.

Scaffolding Area (m^2)^	Average Number of Items/Damaged Items												Partial Hazard Coefficient $W_{s}$
	Frames		Working Platforms		Handrails		Toe Boards		Vertical Elements/Ladders		All Elements		
30–300	1.23	36.64	3.60	28.44	4.45	72.29	1.24	26.73	0.87	6.05	11.45	170.15	0.07
300–600	3,18	75.75	9.14	64.61	8.21	152.54	3.86	47.43	0.89	8.54	25.29	348.86	0.07
600–900	3.64	148.52	9.44	120.88	10.88	355.12	1.72	127.52	1.60	15.12	27.28	767.16	0.04
900–1500	8.08	229.25	11.42	197.67	12.83	657.83	6.00	213.00	1.50	20.83	39.83	1318.58	0.03

**Table 10 ijerph-17-05773-t010:** Foundations with regards to the scaffolding surface area.

Scaffolding Area (*m^2^*)	Average Number of Vertical Elements of All Frames and Foundation pairs	Average Number of Incorrectly Placed Frames and Damaged Underlays	Partial Hazard Coefficient $W_{p}$
30–300	6.02	3.44	0.57
300–600	11.04	5.52	0.50
600–900	12.60	6.36	0.50
900–1500	14.00	6.56	0.47

**Table 11 ijerph-17-05773-t011:** Assessment form for the state of threat of working on scaffolding—case study.

Assessment Form for the State of Threat of Working on Scaffolding			
Technical Factors			
**Scaffolding Type**	**Partial Hazard Coefficient (** $W_{z}$ **)**		**User Selection (Value)**
System	0.16	0.33	0.33
Frame	0.33		
Modular	0.02		
Tubular	0.03		
Carriageable	0.12		
Warsaw	0.34		
Suspended	0.01		
**Damage**	**Number (Value)**	**Partial Hazard Coefficient (** $W_{s}$ **)**	**User Selection (Value)**
The total number of frames	234	If it is not possible to assess the technical conditions of the listed elements, use the designated values: - for scaffolding with an area from 30 to 600 m^2^: $W_{s}$ = 0.07, - for scaffolding with an area from 600 to 900 m^2^: $W_{s}$ = 0.04, - for scaffolding with an area above 900 m^2^: $W_{s}$= 0.03	0.01
The number of damaged frames	1		
The total number of working platforms	204		
The number of damaged working platforms	3		
The total number of handrails	856		
The number of damaged handrails	16		
The total number of toe boards	228		
The number of damaged toe boards	1		
The total number of communication segments, ladders	12		
The number of damaged communication segments, ladders	0		
Total number of elements	1534	0.01	
Total number of all damaged elements	21		
**Foundations**	**Number (Value)**	**Partial Hazard Coefficient (** $W_{p}$ **)**	**User Selection (Value)**
The number of vertical elements in all frames (foundation pairs)	18	If it is not possible to assess the correctness of foundations, use the designated values: - for scaffolding with an area from 30 to 300 m^2^: $W_{p}=0.57$, - for scaffolding with an area from 300 to 900 m^2^: $W_{p}=0.50$ - for scaffolding with an area above 900 m^2^: $W_{p}=0.47$	0.17
The number of damaged underlays	3	0.17	
**Organizational factors**			
**Type of Work Carried Out at a Construction Site**	**Partial Hazard Coefficient (** $W_{r}$ **)**		**User Selection (Value)**
Construction work	0.50	0.50	0.50
Renovation work	0.38		
Scaffolding assembly	0.06		
Scaffolding dismantling	0.06		
**Height at Which Work Will be Carried Out on a Construction site**	**Partial Hazard Coefficient (** $W_{h}$ **)**		**User Selection (Value)**
less than 1 m	0.04	0.04	1.00
1–2 m	0.08	0.08	
3–4 m	0.37	0.37	
5–6 m	0.19	0.19	
7–8 m	0.11	0.11	
9–10 m	0.07	0.07	
11–12 m	0.05	0.05	
more than 12 m	0.10	0.10	
**Time When Work Will be Carried on Scaffolding**	**Partial Hazard Coefficient (** $W_{g}$ **)**		**User Selection (Value)**
07.00–07:59	0.06	0.06	0.96
08:00–08:59	0.10	0.10	
09:00–09:59	0.14	0.14	
10:00–10:59	0.09	0.09	
11:00–11:59	0.09	0.09	
12:00–12:59	0.10	0.10	
13:00–13:59	0.07	0.07	
14:00–14:59	0.15	0.15	
15:00–15:59	0.10	0.10	
16:00–16:59	0.06	0.06	
17:00–17:59	0.02		
18:00–06:59	0.02		
**Enterprise Size—the Number of Persons Employed at Construction Site**	**Partial Hazard Coefficient (** $W_{f}$ **)**		**User Selection (Value)**
Micro-enterprises (1–9)	0.59	0.36	0.36
Small enterprises (10–49)	0.36		
Medium enterprises (50–249)	0.05		
Big enterprises (above 250)	0.01		
**Human Factors**			
**Age of People Employed to Work on Scaffolding**	**Partial Hazard Coefficient (** $W_{w}$ **)**	**Number of Employees (Value)**	**User Selection (Value)**
18–19	0.02		0.22
20–29	0.22	5	
30–39	0.20	6	
40–49	0.24	4	
50–59	0.28		
>60	0.04		
	Total number of employees	15	
**Work Experience of People Employed to Work on Scaffolding**	**Partial Hazard Coefficient (** $W_{d}$ **)**	**Number of Employees (Value)**	**User Selection (Value)**
1 year and less	0.59	5	0.24
2 to 3	0.17		
4 to 5	0.09		
6 to 10	0.09	6	
11 to 15	0.02		
16 to 20	0.02	4	
more than 20 years	0.02		
	Total number of employees	15	
**Employment Status (Form of Employment)**	**Partial Hazard Coefficient (** $W_{z}$ **)**	**Number of Employees (Value)**	**User Selection (Value)**
Self-employed	0.03		0.28
Worker employed for an indefinite period, full-time job	0.28	15	
Worker employed on fixed-term, full-time job	0.43		
Trainee/student	0.01		
Contract for specific work	0.08		
Contract of mandate	0.17		
	Total number of employees	15	
**The simplified coefficient for assessing the state of threat of working on scaffolding** $UW_{z}$			4.07
**The exact coefficient for assessing the state of threat of working on scaffolding** $\;W_{z}$			1.68

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
