# Peer review of "Quantitative Assessment of the State of Threat of Working on Construction Scaffolding"

_ijerph, 2020, doi:10.3390/ijerph17165773_

Round 1
Reviewer 1 Report
I thought that this manuscript had some real merit. It tackles an interesting subject matter, and I think an important one as well. I am a little surprised that this is not being presented to a construction management journal (of which there are several), but I suppose that there is a connection with IJERPH readership here.
The manuscript is fairly well written but has a number of English grammar and sentence structure issues in them that would probably best be handled through the use of an English-speaking copy editor. Here are some of the suggestions, although they became too numerous to count after a while, so I abandoned the effort.
Line 47 - replace with “…tower structures, and chimneys”
Line 57 - replace with “…and modernization of existing buildings”
Line 67 - replace with “…performed through analysis”
Line 69 - replace with “…employees”
Line 80 - please re-write the sentence starting “In the absence of…” It does not sound like a real sentence.
Line 81 - change to “method”
Line 84 - please address the phrase “type of assessed the work process”
Line 86 - should these be all initial caps in “Job safety analysis”?
Line 91 - please address the phrase “the both of this method”
Line 92 - please address the phrase “involved in a work”…in fact, the whole sentence could use some work.
Line 95 - replace with “use a checklist”.
Lines 95-96 - please address the phrase “all this methods”.
I think that much of the readership may not know what a “voivodship” is since it seems to have a strictly Polish meaning. Please elaborate further on this.
The audience might be better served by knowing what “Warsaw-type scaffolding” is relative to other scaffolding, possibly even through some form of visual. The description from lines 231-239 suggest that some visual representation of these types of scaffolding might be helpful.
You might want to adjust some of your figures in column 1 of Table 3. It looks like the < sign is supposed to be the open parentheses sign. This happens in other tables as well.
On page 10, Line 61, I’m not entirely sure what “contraindication” means….you might want to clarify.
In Table 11, under “foundations” in Column 1, I think that one of the responses there should say “damaged underlays”.
Two of the most interesting findings from this research was the timing of the accidents and the staggering amount of data showing that people on the job for 1 year or less find themselves injured on the job. I was a little surprised to see that this was not addressed more in the Discussion section. For example, the first finding might suggest that a staggered work schedule might eliminate some of the lackadaisical nature regarding timing of accident (MIGHT!). Also, the second finding suggests a real need for more training before entering the construction workforce. This would seem like a good opening to talk about the normal procedures for a Polish work force.
Author Response
Dear Reviewer,
Thank you very much for your review and your critical comment, which allow to do our article better. We apologize that the previous version of our paper did not meet your expectations. We hope that in the current version of the paper, we have taken into account all your critical remarks. We also hope, that the current version meets your expectations.
Comment: I thought that this manuscript had some real merit. It tackles an interesting subject matter, and I think an important one as well. I am a little surprised that this is not being presented to a construction management journal (of which there are several), but I suppose that there is a connection with IJERPH readership here.
The manuscript is fairly well written but has a number of English grammar and sentence structure issues in them that would probably best be handled through the use of an English-speaking copy editor. Here are some of the suggestions, although they became too numerous to count after a while, so I abandoned the effort.
Line 47 - replace with “…tower structures, and chimneys”
Line 57 - replace with “…and modernization of existing buildings”
Line 67 - replace with “…performed through analysis”
Line 69 - replace with “…employees”
Line 80 - please re-write the sentence starting “In the absence of…” It does not sound like a real sentence.
Line 81 - change to “method”
Line 84 - please address the phrase “type of assessed the work process”
Line 86 - should these be all initial caps in “Job safety analysis”?
Line 91 - please address the phrase “the both of this method”
Line 92 - please address the phrase “involved in a work”…in fact, the whole sentence could use some work.
Line 95 - replace with “use a checklist”.
Lines 95-96 - please address the phrase “all this methods”.
Answer: All the comments included in the review have been provide during the correction of the article. Our manuscript had checked by a native English speaking from Department of Foreign Languages from Wroclaw University of Science and Technology. We hope that now the English is correct and readable.
Comment: I think that much of the readership may not know what a “voivodship” is since it seems to have a strictly Polish meaning. Please elaborate further on this.
Answer: We agree with your comment. At the present version of this paper we added definition of “voivodship”, as follows:
The voivodeship is the highest-level administrative subdivision of Poland, corresponding to a "province" in many other countries [29].
[29] Hoła, B.; Nowobilski, T. Classification of economic regions with regards to selected factors characterizing the construction industry. Sustain. 2018, 10, 1637.
Comment: The audience might be better served by knowing what “Warsaw-type scaffolding” is relative to other scaffolding, possibly even through some form of visual. The description from lines 231-239 suggest that some visual representation of these types of scaffolding might be helpful.
Answer: Thank you very much for your comment. We added more information about the Warsaw-type scaffolding and the figure 1.
Comment: You might want to adjust some of your figures in column 1 of Table 3. It looks like the < sign is supposed to be the open parentheses sign. This happens in other tables as well.
Answer: We agree with the reviewer's comments. At the present version of this paper we improved it.
Comment: On page 10, Line 61, I’m not entirely sure what “contraindication” means….you might want to clarify.
Answer: We added information about exemplary contraindications, as follows:
“Unfortunately, construction practice shows that people employed under civil law contracts are not properly prepared to perform work, i.e. they did not undergo general and on-the-job trainings, did not have medical examinations, and do not possess current medical certificates concerning the absence of contraindications to work on a specific job position, e.g. an employee is prohibited from lifting loads over 5 kg or working at heights, etc. ”
Comment: In Table 11, under “foundations” in Column 1, I think that one of the responses there should say “damaged underlays”.
Answer: We agree with the reviewer's comments and we changed it.
Comment: Two of the most interesting findings from this research was the timing of the accidents and the staggering amount of data showing that people on the job for 1 year or less find themselves injured on the job. I was a little surprised to see that this was not addressed more in the Discussion section. For example, the first finding might suggest that a staggered work schedule might eliminate some of the lackadaisical nature regarding timing of accident (MIGHT!). Also, the second finding suggests a real need for more training before entering the construction workforce. This would seem like a good opening to talk about the normal procedures for a Polish work force.
Answer: We agree with the reviewer's comments. The conclusions indicated by the Reviewer about "the timing of the accidents and the data showing that people on the job for 1 year or less find themselves injured on the job" are the subject of detailed analyzes currently conducted by the Authors. In subsequent works, the Reviewer suggestion will be discussed in detail and supported by the analyzes.

Reviewer 2 Report
This paper presents the quantitative assessment on the state of threat to working on construction scaffolding based on the ten proposed parameters concerning the possibility of occupational accidents classified into three groups of influencing factors including technical, organizational and human factors. Also, a test method is proposed with numerical scales for assessing the state of threat to working on construction scaffolding. The presented results are genuine, initiative and practically useful, in particular for assessing and improving the safety of employees working on scaffolding at height. The conclusions are inclusive, sound and convincing. The paper is well organised and written, including table and figures. The paper provides much new information so it is worthwhile to publish. There are some technical, issues which need to be further addressed and numerous editorial and grammatical errors which need to be further revised before the paper can be published. Also the formats for some references are wrong and need to be revised. These issues are clearly marked in the PDF version of the paper. The authors should pay attention to them.

Author Response
Dear Reviewer,
Thank you very much for your review and your critical comment, which allow to do our article better. We apologize that the previous version of our paper did not meet your expectations. We hope that in the current version of the paper, we have taken into account all your critical remarks. We also hope, that the current version meets your expectations.
Our manuscript had checked by a native English speaking from Department of Foreign Languages from Wroclaw University of Science and Technology. We hope that now the English is correct and readable.
Comment: This paper presents the quantitative assessment on the state of threat to working on construction scaffolding based on the ten proposed parameters concerning the possibility of occupational accidents classified into three groups of influencing factors including technical, organizational and human factors. Also, a test method is proposed with numerical scales for assessing the state of threat to working on construction scaffolding. The presented results are genuine, initiative and practically useful, in particular for assessing and improving the safety of employees working on scaffolding at height. The conclusions are inclusive, sound and convincing. The paper is well organised and written, including table and figures. The paper provides much new information so it is worthwhile to publish. There are some technical, issues which need to be further addressed and numerous editorial and grammatical errors which need to be further revised before the paper can be published. Also the formats for some references are wrong and need to be revised. These issues are clearly marked in the PDF version of the paper. The authors should pay attention to them.
Answer: All the comments included in the review have been provide during the correction of the article.
